# Automatic Sleep Staging Using BiRNN with Data Augmentation and Label Redirection

**Yulin Gong** [1,*] , **Fatong Wang** [1], **Yudan Lv** [2], **Chang Liu** [2] and **Tianxing Li** [1]

1   School of Electronic Information, Changchun University of Science and Technology, Changchun 130022, China; fatong1227@163.com (F.W.); tianxing_l@163.com (T.L.)
2   The Department of Neurology, Jilin University, Changchun 130012, China; lvyudan@sina.com (Y.L.); drliuchang@jlu.edu.cn (C.L.)
*   Correspondence: gongyulin@cust.edu.cn; Tel.: +86-135-1446-1696

**Abstract:** Sleep staging has always been a hot topic in the field of sleep medicine, and it is the cornerstone of research on sleep problems. At present, sleep staging heavily relies on manual interpretation, which is a time-consuming and laborious task with subjective interpretation factors. In this paper, we propose an automatic sleep stage classification model based on the Bidirectional Recurrent Neural Network (BiRNN) with data bundling augmentation and label redirection for accurate sleep staging. Through extensive analysis, we discovered that the incorrect classification labels are primarily concentrated in the transition and nonrapid eye movement stage I (N1). Therefore, our model utilizes a sliding window input to enhance data bundling and an attention mechanism to improve feature enhancement after label redirection. This approach focuses on mining latent features during the N1 and transition periods, which can further improve the network model's classification performance. We evaluated on multiple public datasets and achieved an overall accuracy rate of 87.3%, with the highest accuracy rate reaching 93.5%. Additionally, the network model's macro F1 score reached 82.5%. Finally, we used the optimal network model to study the impact of different EEG channels on the accuracy of each sleep stage.

**Keywords:** sleep staging; data augmentation; label redirection; physiological signal; BiRNN





## 1. Introduction

Sleep is one of the most important activities in a person's life, and all life activities are inseparable from the process of sleep [1]. In 2019, the COVID-19 pandemic swept across the world. A new study published in the authoritative medical journal *The Lancet Respiratory Medicine* showed that more than half of the patients who were infected with COVID-19 early on still had lingering sequelae, and 31% of them reported difficulty sleeping [2]. The diagnosis of sleep problems cannot be separated from important data—the sleep-staging chart. The earliest sleep-staging chart divided the whole sleep process into six stages: wake stage (W), N1, nonrapid eye movement stage II (N2), nonrapid eye movement stage III (N3), nonrapid eye movement stage IV (N4), and rapid eye movement stage (REM) according to the R&K standard [3]. In 2007, the American Academy of Sleep Medicine (AASM) merged the N3 stage and N4 stage of the R&K standard into the N3 stage and released the Manual for the Scoring of Sleep and Associated Events [4]. The 5-class sleep-staging chart with 30 s data as a window became the mainstream and the scoring standard in the field of sleep medicine. Today, the acquisition of sleep-staging charts in all hospitals still relies on the original manual interpretation, which not only consumes a lot of manpower and material resources, but also is prone to errors due to personal subjective factors or fatigue [5]. The mainstream machine learning classifiers include support vector machine (SVM), random forest, relevance vector machine (RVM), K-nearest neighbor (KNN), hidden Markov model, and Back Propagation (BP) network; deep learning classifiers include Convolutional Neural Networks (CNNs), Recurrent Neural Network (RNN), Artificial Neural Network (ANN),

and combinations of different neural networks. Although machine learning and deep learning have achieved remarkable results in the field of sleep staging, there are still the following problems in automatic sleep staging by algorithm:

1. Misclassifying a certain stage as the most common category in the whole due to sample data imbalance [6];
2. The recognition accuracy of the N1 stage between the W stage and N2 stage is generally lower than 50%;
3. There is a large difference among databases [7], and the channel mismatch problem is significant.

To solve the above problems, we propose a sleep stage automatic classification model based on a dual-layer BiRNN network with data bundling enhancement and label redirection, which is expected to replace the manual division of sleep stages. The model combines a data augmentation algorithm and an attention mechanism to realize the automatic scoring of sleep stages. The main contributions of this work are as follows:

- We build a sleep stage automatic classification model based on a dual-layer BiRNN network with data bundling enhancement and label redirection. By combining an attention mechanism to realize feature enhancement after label redirection, we enhance the feature distinguishability of the sleep transition period and the N1 stage and further improve the classification performance of the network model;
- We design a signal bundling enhancement method, which obtains new signal patterns, while preserving long-term context information and reducing the probability of transition stage recognition error;
- We conduct extensive experiments on three public datasets, and the experimental results show that our model improves the expression ability of sleep features by mining deep features of the N1 stage, preliminarily solves the problem of low recognition accuracy of the N1 stage easy-to-recognize as the adjacent stage, and further improves the classification accuracy of the sleep stages.

## 2. Related Work

Sleep stage differentiation, as a typical time series analysis problem, has attracted the attention of many researchers. In recent years, with the continuous expansion of sleep databases, more and more researchers have begun to look for breakthroughs in automatic sleep staging based on deep learning algorithms. At present, mainstream deep learning algorithms, such as CNN, RNN, and Generative Adversarial Network (GAN), have been widely used for automatic sleep staging, especially the sequence network model based on CNN for extracting time-invariant features and frequency information, which has achieved good annotation effects by capturing the correlation between sleep time and scoring. Some researchers proposed a hierarchical RNN named SeqSleepNet [8], which treats this task as a sequence-to-sequence classification task. Meanwhile, hybrid models are also adopted by some researchers. DeepSleepNet [9] uses CNN to extract time-invariant features and uses Bidirectional Long Short-Term Memory (BiLSTM) to learn the transition rules between sleep stages, achieving relatively good classification results.

RNN networks, with their innate advantages in processing temporal signals, have been favored by many researchers. XSleepNet [10] proposed a sequence-to-sequence network staging model based on RNN, which effectively learned features through time-frequency images, and the evaluation results of the model showed that it was at an advanced level in the field of automatic sleep staging. However, the current automatic sleep staging mainly faces three major problems: data imbalance, low accuracy of the N1 stage, and easy errors in the transition stage [6,7]. Data imbalance caused by different sleep stage durations and other reasons is an important factor affecting the accuracy of automatic sleep staging. The low accuracy of the N1 stage compared with other sleep stages is also a problem that many staging methods have not solved.

In this paper, we propose a novel RNN-based network model, which has two main parts: data augmentation and label redirection. In the process of network training, we

effectively capture the salient features of the N1 stage and transition stage and further enhance the correlation between adjacent sleep stages. In addition, we adopt the strategy of sacrificing some redundant data to achieve data balance, thus improving the accuracy of the network model.

In sleep network model research, it is very challenging to achieve accurate sleep stage classification with as few physiological signals as possible. In the current trend of automatic sleep-staging research, most researchers tend to use more electroencephalogram (EEG) channels to capture more sleep-related features to improve the accuracy of automatic staging, although significant results have been achieved [11], but considering the applications of related fields, such as sleep monitoring, it is a trend to use fewer signals to achieve accurate automatic sleep staging. Therefore, we only used single-channel EEG, electrooculography (EOG), and electromyogram (EMG) to conduct automatic sleep-staging research.

## 3. Method

### 3.1. Signal Preprocessing

The polysomnography (PSG) signals highly related to sleep mainly include EEG, EOG, and EMG [12], and professional sleep physicians also identify the characteristic waves of each sleep stage based on these three signals to complete the manual sleep stage annotation. Introducing multiple signals for sleep-staging research can extract more sleep-related features [10] to improve the accuracy of sleep staging, so we used the combination of EEG, EOG, and EMG signals for sleep-staging research. The sampling rates of different signals in the same dataset were mostly different. In order to improve the compatibility of the input network data shape, improve the data consistency, reduce the computation, and prevent overfitting [7], we downsampled the signals above 100 Hz to 100 Hz and upsampled the signals below 50 Hz to 100 Hz, so that they can continuously enhance the useful signals in the training iteration process. Some datasets contained a large amount of signals before turning off the light. If a large amount of awake-state data are introduced in the network training stage, it will lead to a decrease in network staging accuracy due to data imbalance and a large amount of noise interference in the monitoring signal before turning off the light. Therefore, we deleted the data from the start of collection to 5 min before turning off the light. The original data contained a lot of noise interference. We used a second-order IIR notch filter to perform 50 Hz/60 Hz power frequency notch filtering on the original signal. In addition, according to the distribution of characteristic waves of different sleep stages, we used zero-phase delay filter to perform 0.3–40 Hz [13] band-pass filtering on EEG and EOG signals.

### 3.2. Signal Enhancement

To address the problem of low recognition accuracy of N1 stage and easy identification of adjacent stages, this paper proposes a signal-bundling enhancement training method. First, each sleep stage label corresponded to 30 s of EEG, EOG, and EMG data. We grouped the adjacent 90 s of data and labeled them as A, B, and C data segments. We scanned the data with a sliding window of 90 s and a sliding length of 30 s, as shown in Figure 1. Figure 2 shows that the current group AB segment data are the same as the previous group BC segment data, and the current group BC segment data are the same as the next group AB segment data, and the current group C segment data are the same as the second group's data segment A. In the network training stage, we used a sliding window to achieve bundling between adjacent data, thus enhancing the current group C segment data.

By analyzing the confusion matrix of various neural network staging results, it is apparent that N1 stage is more likely to be identified as W stage and N2 stage because it is in the transition stage between W stage and N2 stage and has no obvious features [14]. We bundled the adjacent 90 s data into a whole by scanning with a sliding window, except for the first and last 60 s data, and the remaining data were scanned 3 times to strengthen the correlation between the previous and next data and the current label, that is, each set of window data were bound with the previous and next 60 s data. In addition, because

N1 stage has a short time in the whole sleep process, there will be a problem of sample imbalance in sleep stage, which will lead to N1 stage being biased toward most other stages, such as W stage and N2 stage [15]. To solve this problem, in the process of scanning data with a sliding window, we re-encoded the data label corresponding to N1 stage as 1 and re-encoded the data of other stages as 0, realizing N1 stage label redirection. In the process of network training, each window data will map two labels, one is the original label, and the other is the redirected label. The data with redirected label 1 will strengthen the data features in the training process, and the data with redirected label 0 will not be processed in the training process.

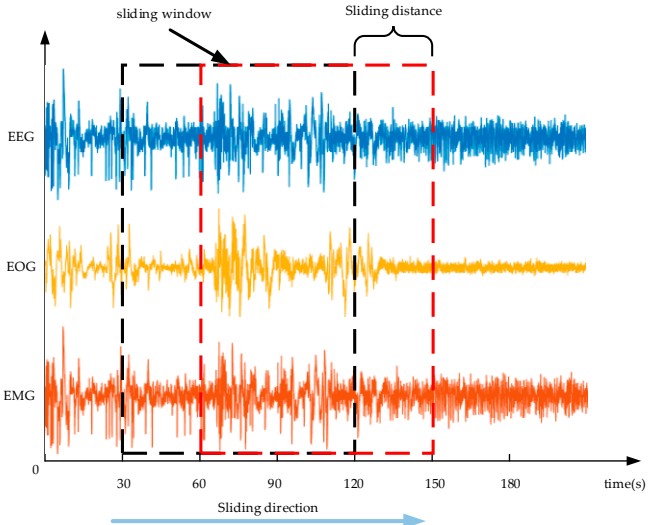

**Figure 1.** Scan of the three processed signals with a sliding window.

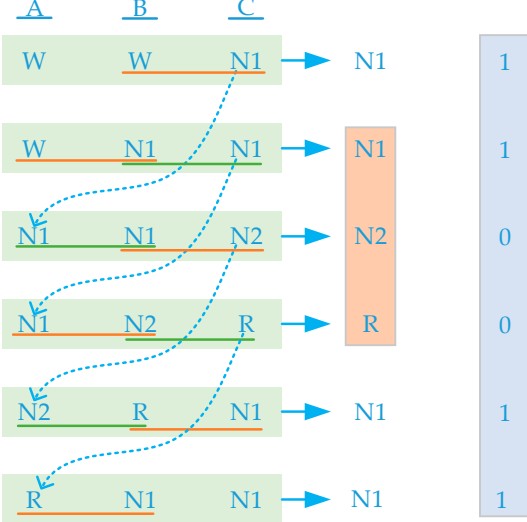

**Figure 2.** Data bundling and label recoding.

### 3.3. BiRNN with Attention Mechanism

In order to extract the salient features related to sleep staging from EEG, EOG, and EMG signals, considering the ability of sleep network to process sequential data and data augmentation, we proposed a dual-layer BiRNN with attention mechanism to learn key features to achieve more accurate sleep-staging results. As shown in Figure 3, the input part of our sleep network consisted of a BiRNN with attention mechanism, which is used for data augmentation and feature extraction. The output part of the sleep network consisted of a bidirectional recurrent neural network with softmax classification layer, which is used

for feature learning and classification. The improved sleep network not only has a stronger ability to capture salient features, but also solves the problem of gradient explosion or disappearance. On the other hand, by proposing a data augmentation method, the feature learning ability of transition period and N1 stage is greatly improved, and the classification performance of sleep network is further optimized.

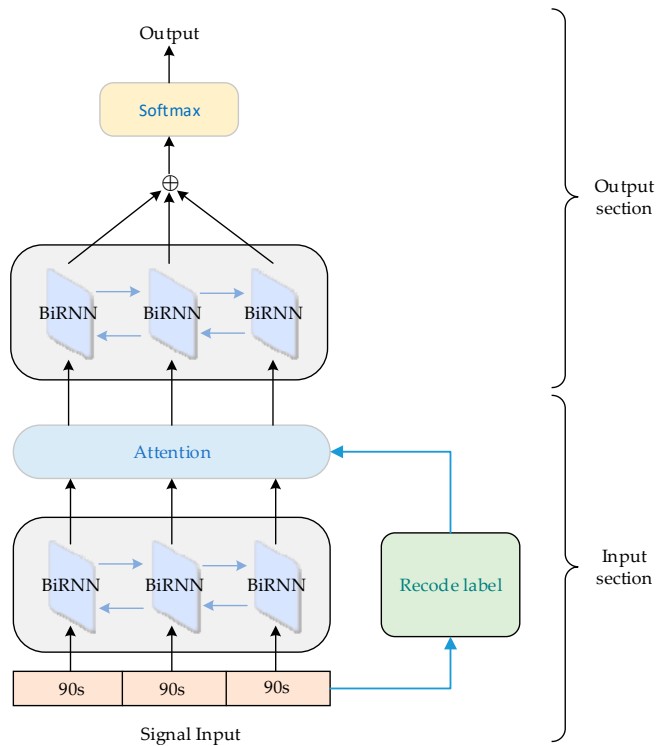

**Figure 3.** Automatic sleep-staging network architecture.

We used the preprocessed data $X \in R^{T \times C}$ as the input of the sleep network, where $T = 90$ represents the data of 90 s for each input, and $C = 3$ represents the number of channels of the input data. Then, we input the data sequence $X$ into the sleep network in the form of sliding windows, where the window size is 90 s and the window sliding distance is 30 s.

First, we used a BiRNN with attention mechanism to extract the sequence features of each window data. The forward and backward recurrent layers of RNN computed the forward sequence $h_t^f$ and backward sequence $h_t^b$ of hidden layer state vectors in opposite directions, defined as follows:

$$h_t^f = \sigma\left(h_{t-1}^f, x_t\right) \tag{1}$$

$$h_t^b = \sigma\left(h_{t-1}^b, x_t\right) \tag{2}$$

$$a_t = W(h_t^b \odot h_t^f) + b_a \tag{3}$$

where $\sigma$ denotes the hidden layer activation function sigma, $a_t$ denotes the output computation, $\odot$ denotes the vector combination, and $b_a$ denotes the bias vectors. We used the sigmoid function as the activation function, which is defined as follows:

$$S(x) = \frac{1}{1 + e^{-x}} \tag{4}$$

We compared Long Short-Term Memory (LSTM) and Gate Recurrent Unit (GRU); GRU is a variant of LSTM, and compared to LSTM, GRU's internal structure is much simpler,

and the performance is almost the same [16]. In order to shorten the training time of the sleep network as much as possible, we finally chose GRU. The purpose of introducing the attention mechanism layer was to reweight the vectors that are redirected to label 1. As shown in Figure 4, on the one hand, it strengthens the attention to the vectors with label 1, and, on the other hand, it further emphasizes the feature vectors that are strongly related to the target. Less attention was paid to the vectors with redirected label 0. The formula for calculating the attention weight at time $t$ is as follows:

$$\alpha_t = \frac{\exp(f(a_t))}{\sum_{i=1}^{T} \exp(f(a_i))} \tag{5}$$

where $f(x)$ is the scoring function of the attention layer, defined as follows:

$$f(x) = a_t(W_m \otimes W_t) \tag{6}$$

where $W_m$ is a trainable weight matrix, $W_t$ is a redirect weight matrix, and $\otimes$ denotes matrix combination.

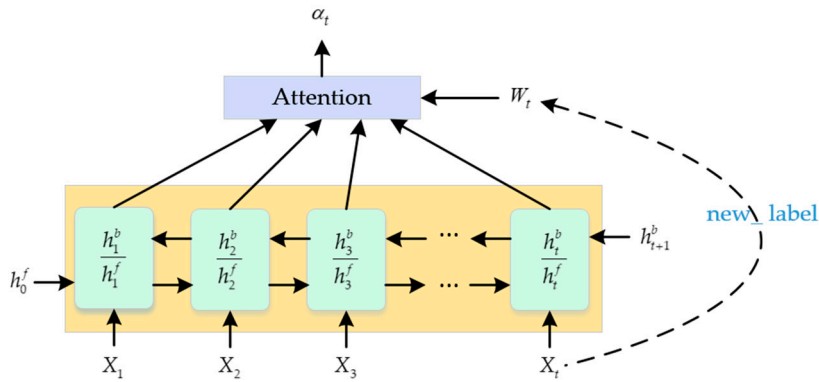

**Figure 4.** Feature enhancement based on attention mechanism.

Like the BiRNN based on attention mechanism, we used the attention feature vector output by the attention layer as the input of the second layer RNN network. The hidden layer also used GRU, and the forward and backward sequences of the hidden layer state vector were calculated in the same way as (1) and (2). The output vector $O$ was calculated as follows:

$$O_t = o_{t-1} \oplus o_t \oplus o_{t+1} \tag{7}$$

Finally, the output vector was fed into the softmax layer to complete the classification, and the output sequence $Y = (y_1, y_2, \ldots, y_p)$ was obtained, where $y_p$ represents the probability distribution of the sleep stage output. We chose the cross-entropy loss function as our loss function, which is commonly used and has relatively good results.

## 4. Experiment

### 4.1. Sleep Database

#### 4.1.1. Sleep-EDF Database Expanded

The dataset used is the public dataset Sleep-EDF Database Expanded collected by PhysioNet website [17]. The database contains 197 whole-night sleep records, including EEG, EOG, chin EMG, and manual annotations of sleep stages. The EEG was collected at Fpz-Cz and Pz-Oz locations. The sampling frequency of EEG and EOG was 100 Hz; the sampling frequency of EMG in the Sleep Cassette (SC) subset was 1 Hz, and the sampling frequency of EMG in the Sleep Telemetry (ST) subset was 100 Hz. Among them, half of the subjects in the ST subset took temazepam and the other half took placebo. Temazepam is clinically used to treat sleep disorders, so only 153 whole-night sleep data points from the SC subset were used in the experiment.

4.1.2. Sleep Heart Health Study (SHHS)

The SHHS is a multicenter cohort study conducted by the National Heart, Lung, and Blood Institute to determine the cardiovascular and other consequences of sleep-disordered breathing [18,19]. The database contains 6441 subjects who participated in sleep-monitoring records, and we only used the data from SHHS1 for experimental testing, because its dataset is large enough. The sampling frequency of both EEGs (C4-A1 and C3-A2) and EMG signals was 125 Hz, while the sampling frequency of EOG signal was 50 Hz.

4.1.3. Haaglanden Medisch Centrum Sleep-Staging Database

The dataset comes from 151 whole-night sleep records collected by Haaglanden Medisch Centrum sleep center [20]. The dataset includes EEG, EOG, and EMG as well as manual annotations of sleep stages by professional sleep technicians. The PSG data included 4 EEGs (F4-M1, C4-M1, O2-M1, and C3-M2), 2 EOGs (E1-M2 and E2-M2), and all signals had a sampling frequency of 256 Hz.

*4.2. Experiment Setup*

In order to investigate the impact of different EEG channels on the staging of N1 sleep stage, we used two EEG channels, Fpz-Cz and Pz-Oz, from the Sleep-EDF Database Expanded as our study subjects. For the SHHS dataset, we selected C4-A1 as our EEG channel of interest, and for the Haaglanden Medisch Centrum sleep-staging database, we chose C4-M1 as our EEG channel of interest.

To better validate the universality of the network model, we tested it on different databases. Because some databases had a small sample size that did not meet the optimal training requirements of the model, we used 30-fold cross-validation to select the model and optimize its parameters in those databases with small sample sizes, and achieved excellent experimental results. During the network model training process, the different probabilities of occurrence of each sleep stage meant that there was a huge imbalance in the sample sizes of each sleep stage, which in turn affected the accuracy of the model training. If the sample size of one sleep stage was much larger than that of other sleep stages, or if the sample size of one sleep stage was much smaller than that of other sleep stages, a serious problem of sample imbalance would cause the final classification result to be biased toward the sleep stage with many samples. This phenomenon is particularly evident in the sleep transition stage [6,15]. To solve this problem, we used a clever method of randomly dropping data in the preprocessing step of sleep network training to remove some continuous, large sample size sleep stage data, while leaving the data in the transition stage untreated. Additionally, we removed data with a large amount of wake time during the lights-on period, which would affect the training effectiveness of the model.

The network model was implemented in tensorflow_gpu_2.4.0 framework; python version was 3.8; computer processor was Intel® Core™ i9-10940X CPU @3.30 GHz (Intel Corporation, Santa Clara, CA, USA), using NVIDIA GeForce RTX 3090 Ti for network training, and learning rate was $10^{-4}$. Without considering the number of cross-validation folds, the training time was significantly shortened compared with other methods [8].

**5. Results and Discussion**

We conducted a study analyzing the results of using single-channel EEG automatic sleep staging. All automatic sleep-staging algorithms, including the most advanced deep learning algorithms, show that the sleep-staging results of the Wake and REM stages in the 5-stage classification are the most accurate, almost reaching clinical levels. However, the accuracy of the N1 stage is much lower than the average accuracy of other stages [21]. Most research results show that the average accuracy of the N1 stage is only about 40% [22], which largely hinders the pace of automatic sleep staging replacing manual sleep staging. On the one hand, the N1 stage is a transitional stage from the Wake to N2 stage, and its significant features are not obvious, which can easily be confused with adjacent stages [14]. Even experienced doctors may make subjective judgment errors. On the other hand, the

N1 stage accounts for a relatively small proportion in the entire sleep process [13]. In the process of network training, due to the imbalance of samples, results that originally belonged to the N1 stage tend to be biased toward those stages with larger sample sizes. We used the SHHS dataset as the test benchmark to compare the automatic sleep-staging results before and after data augmentation. Table 1 shows the confusion matrix obtained using the attention-based RNN. Through analysis of the confusion matrix, the same low accuracy of the N1 sleep stage was observed, and the misclassification results were concentrated in the adjacent sleep stages. Similar issues were also present in other sleep stages, but not as severe as the N1 stage. Table 2 shows the confusion matrix obtained after data augmentation. The main diagonal of each confusion matrix represents true (TP) values, indicating the number of stages correctly classified. By comparing the confusion matrices, the classification effect of the N1 stage was greatly improved, and the probability of misclassification between the adjacent sleep stages was significantly decreased; especially the effect of the N1 stage is the most significant. This is because we enhanced the network's attention to the N1 stage during label redirection. The results show that the accuracy of the Wake stage increased by 7.48%, the N1 stage increased by 14.94%, the N3 stage increased by 2.56%, and the REM stage increased by 0.21%. After introducing data-bundling augmentation and label redirection, the network model performed better than before and was superior to most automatic sleep-staging models in terms of the N1 stage classification performance.

**Table 1.** Confusion matrix without using data augmentation.

|  | **Wake** | **N1** | **N2** | **N3** | **REM** |
|---|---|---|---|---|---|
| Wake | 81.5% (3675) | 8.94% (403) | 5.86% (264) | 0.44% (20) | 3.26% (147) |
| N1 | 19.19% (530) | 44.57% (1231) | 30.41% (840) | 0.18% (5) | 5.65% (156) |
| N2 | 2.78% (205) | 6.34% (468) | 78.09% (5761) | 7.95% (586) | 4.84% (357) |
| N3 | 1.74% (97) | 0.07% (4) | 13.84% (774) | 84.24% (4710) | 0.11% (6) |
| REM | 0.62% (47) | 2.54% (193) | 4.74% (361) | 0.14% (11) | 91.96% (6999) |

**Table 2.** Confusion matrix with data augmentation.

|  | **Wake** | **N1** | **N2** | **N3** | **REM** |
|---|---|---|---|---|---|
| Wake | 88.98% (4012) | 6.61% (298) | 3.10% (140) | 0.38% (17) | 0.93% (42) |
| N1 | 14.93% (412) | 59.51% (1644) | 21.73% (600) | 0.36% (10) | 3.47% (96) |
| N2 | 3.47% (256) | 6.20% (457) | 76.98% (5679) | 8.20% (605) | 5.15% (380) |
| N3 | 0.77% (43) | 0.09% (5) | 12.30% (688) | 86.80% (4853) | 0.04% (2) |
| REM | 0.41% (31) | 2.48% (189) | 4.89% (372) | 0.05% (4) | 92.17% (7015) |

Through the experimental comparison of the radar chart in Figure 5 and the random tester experiment results shown in Figure 6, we can draw the following conclusions more intuitively:

1. Throughout the sleep process, the N1 stage has the shortest total duration proportion, while the N2 stage has the longest total duration proportion. Therefore, attention to the problem of imbalanced training data is essential during the network model training process;

2.  Although the accuracy of the N1 stage was greatly improved in our network, the importance of N1 stage data for clinical diagnosis cannot be ignored. Further breakthroughs are needed to meet clinical application standards;

3.  Our network model provides reference values for the classification performance of Wake stage and N1 stage research.

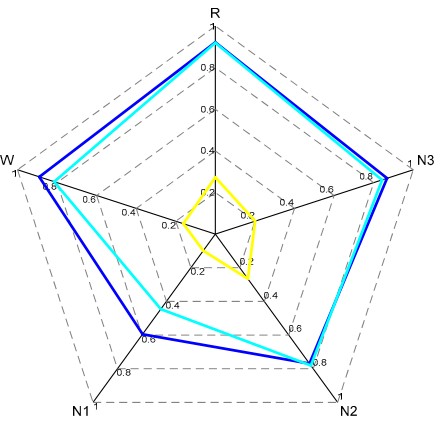

**Figure 5.** The yellow radar chart represents the time distribution of each sleep stage; the cyan radar chart represents the distribution of accuracy for each sleep stage without using data augmentation and label redirection, and the blue radar chart represents the distribution of accuracy for each sleep stage obtained by testing our model, corresponding to the confusion matrix in Table 1.

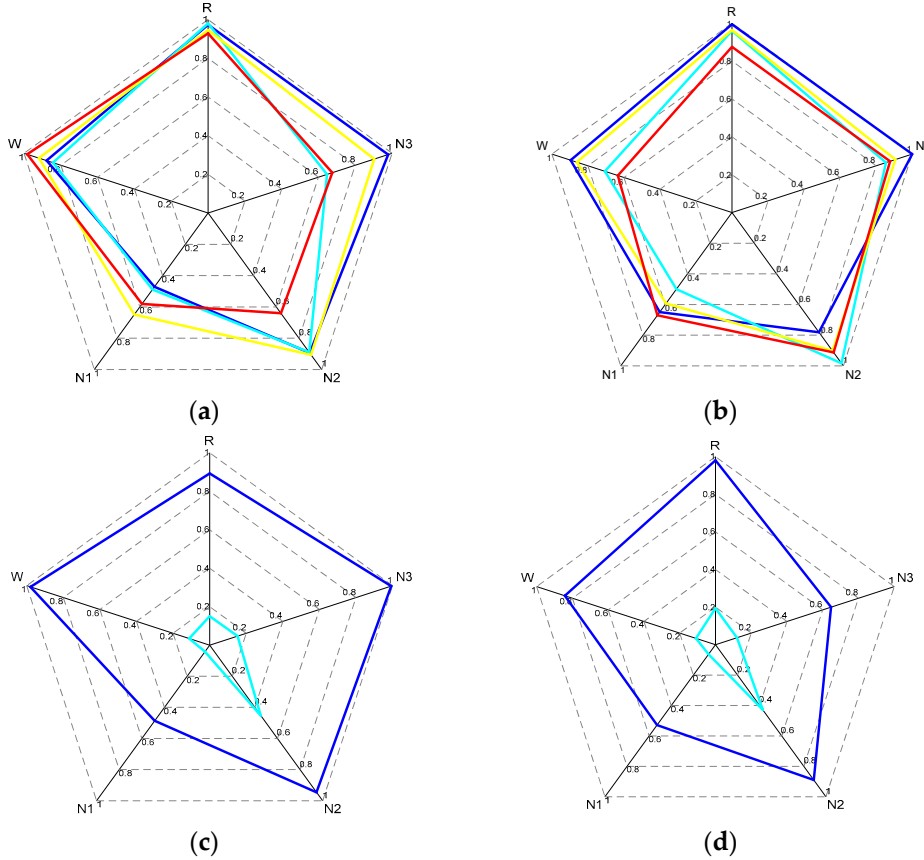

**Figure 6.** Radar plots (**a**,**b**) show the sleep stage classification accuracies for 8 randomly selected subjects from the database. Radar plots (**c**,**d**) show the sleep stage accuracies and proportions of sleep time for 2 randomly selected subjects from the SHHS database, where blue represents accuracy and cyan represents the proportion of sleep time for each stage.

To further validate the performance of the proposed sleep-staging network, we tested it on three publicly available datasets. Because the SHHS dataset contains a large amount of subject data and our staging network received sufficient training on this dataset, we achieved the best staging results on this dataset. As shown in Figure 7a, the staging accuracy obtained using the SHHS dataset reached 93.5%. If the classification results of the N1 stage are not considered, the model's best performance is comparable to manually staged results after sufficient training and calibration. By comparing the manually annotated staging and the network output staging error graph, it is apparent that the network is more likely to make errors in the N1 stage and the transitional stage. Through analysis of the dataset, we found that the N1 stage has the least amount of data and the most difficult distinction process, and about 40% of the data are in the transitional stage, which often contains multiple sleep stages, making it very challenging to classify [14]. However, our sleep-staging network achieved some results in these two aspects. As shown in Figure 7b,c, our network model also shows the best performance on the Sleep-EDF Database Expanded and Haaglanden Medisch Centrum sleep-staging datasets. The F1 scores for these two datasets were 79.6.3 ± 3.6%, slightly lower than the SHHS dataset's testing results. The overall accuracy test results were the highest for the SHHS dataset, followed by the Sleep-EDF Database Expanded dataset and the Haaglanden Medisch Centrum sleep-staging dataset, with accuracies of 87.3%, 86.2%, and 85.9%, respectively. In addition, redirecting the attention weights toward the N1 stage through label redirection showed that the attention layer can better capture the typical features of the N1 stage and reduce the probability of misclassifying the N1 stage as the Wake or N2 stage.

Table 3 presents a comparison of the various network performances using Fpz-Cz EEG data, along with EOG and EMG data, as the study subject in the Sleep-EDF Database Expanded. The table shows the predictive performance (accuracy) of each network model for each stage of sleep, as well as the overall staging accuracy and F1 score. As shown in the table, the overall accuracy of most models is below 82%, while our model achieves a staging accuracy of 86.2% and an MF1 of 80.5%. Our proposed model outperforms other network models in processing the N1 stage, indicating that using data augmentation and label redirection is more conducive to mining features that are easy to distinguish from adjacent sleep stages. Our proposed model also shows good performance in classifying other sleep stages and is on par with other advanced network models, especially in REM stage classification where it exhibits the best performance.

**Table 3.** Comparison of our proposed automatic sleep-staging algorithm with other advanced algorithms using Sleep-EDF Database Expanded as the reference standard. (ACC means overall accuracy, and MF1 means macro F1 score).

|  | Wake | N1 | N2 | N3 | REM | ACC | MF1 |
|---|---|---|---|---|---|---|---|
| Proposed Network | 89.7 | 54.5 | 76.9 | 86.8 | 87.2 | 86.2 | 80.5 |
| Baseline RNN | 89.9 | 43.6 | 55.6 | 77.5 | 80.4 | 77.6 | 73.3 |
| ARNN | 90.7 | 44.3 | 32.4 | 73.5 | 76.5 | 80.1 | 73.6 |
| 1-max CNN [23] | 77.4 | 40.6 | 87.4 | 86.0 | 82.3 | 82.3 | 73.8 |
| Dense Encoder [24] | 88.5 | 41.9 | 86.0 | 81.1 | 84.0 | 82.1 | 76.3 |
| IITNet [25] | 87.7 | 43.4 | 87.7 | 86.7 | 82.5 | 83.9 | 77.6 |
| SAE [14] | 71.6 | 47.0 | 84.6 | 84.0 | 81.4 | 78.9 | 73.7 |
| ResnetMHA [26] | 90.2 | 48.3 | 87.8 | 85.6 | 83.0 | 84.3 | 79.0 |
| FCNN+RNN [10] | 92.5 | 47.3 | 85.0 | 79.2 | 78.9 | 82.8 | 76.6 |
| SeqSleepNet [8] | 92.2 | 47.8 | 84.9 | 77.2 | 79.9 | 82.6 | 76.4 |
| AttnSleep [27] | 92.0 | 42.0 | 85.0 | 82.1 | 74.1 | 81.3 | 75.1 |
| SleepTransformer [28] | 91.7 | 40.4 | 84.3 | 77.9 | 77.2 | 81.4 | 74.3 |
| Deepsleepnet [9] | 84.7 | 46.6 | 85.9 | 84.8 | 82.4 | 82.0 | 76.9 |

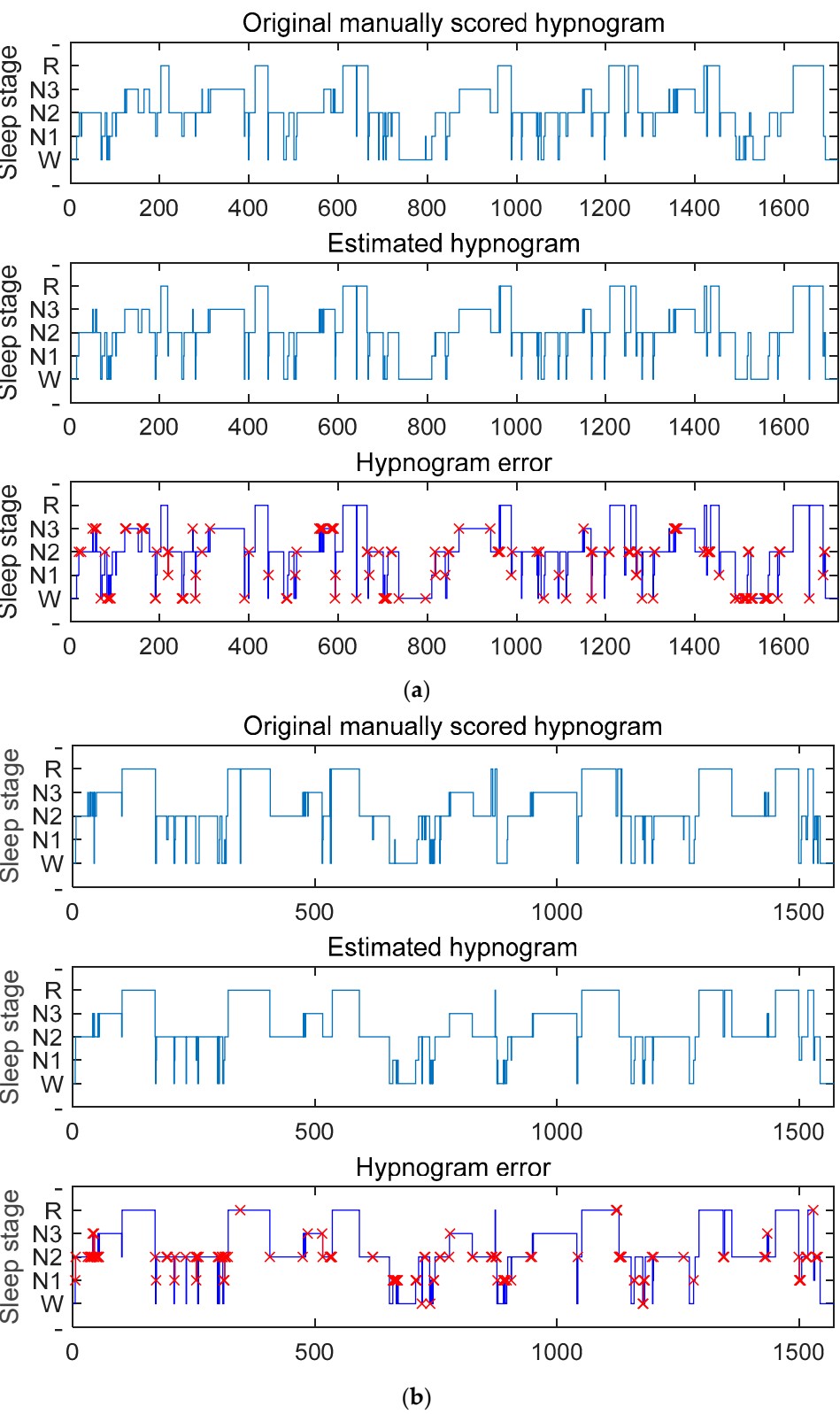

(**a**)

(**b**)

**Figure 7.** *Cont.*

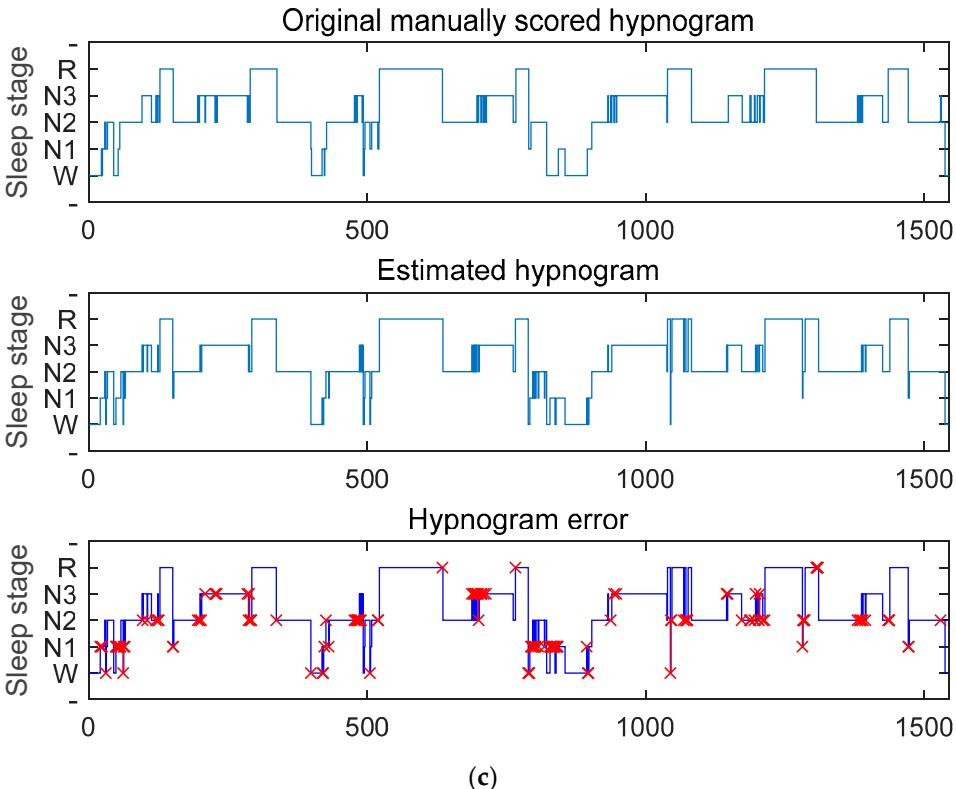

(**c**)

**Figure 7.** The results of testing with different datasets are shown in (**a**) SHHS dataset, (**b**) Sleep-EDF Database Expanded, and (**c**) Haaglanden Medisch Centrum sleep-staging dataset. The figures include the original manually scored sleep graph (**top**), the estimated sleep graph using our algorithm (**middle**), and the error sleep graph comparison between the two (**bottom**), with red x indicating misclassification.

In addition, we investigated the impact of different EEG channels on the classification of sleep stages. We selected the Fpz-Cz and Pz-Oz EEG channels from the Sleep-EDF Database Expanded as the study objects. To reduce the computational complexity of the automatic sleep network, current mainstream automatic sleep-staging research mostly uses single-channel EEG research methods. However, a summary of previous research results shows that using multichannel EEG achieves higher accuracy in automatic sleep staging than using only single-channel EEG [11,29,30]. Sleep is a complex physiological process, and different brain regions are active to varying degrees during different stages of sleep. Extracting EEG features highly correlated with sleep stages from those active brain regions can more accurately classify sleep stages. Figure 8 shows a comparison of results for 20 randomly selected testers using the best network model. From the figure, it is evident that the accuracy of the Wake period, N2 period, and N3 period is not greatly affected by using the Fpz-Cz or Pz-Oz EEG channels, with individual testers showing significant differences, which may be related to the quality of the tester's EEG signal. For the N1 period and REM period, using the Fpz-Cz EEG channel performed better than using the Pz-Oz EEG channel. In summary, different EEG channels have varying contributions to the classification of sleep stages. Considering the limited sleep information contained in single-channel EEG and the difficulty in improving the classification accuracy of the N1 stage, the use of a multichannel EEG combination is of great research significance in overcoming this bottleneck.

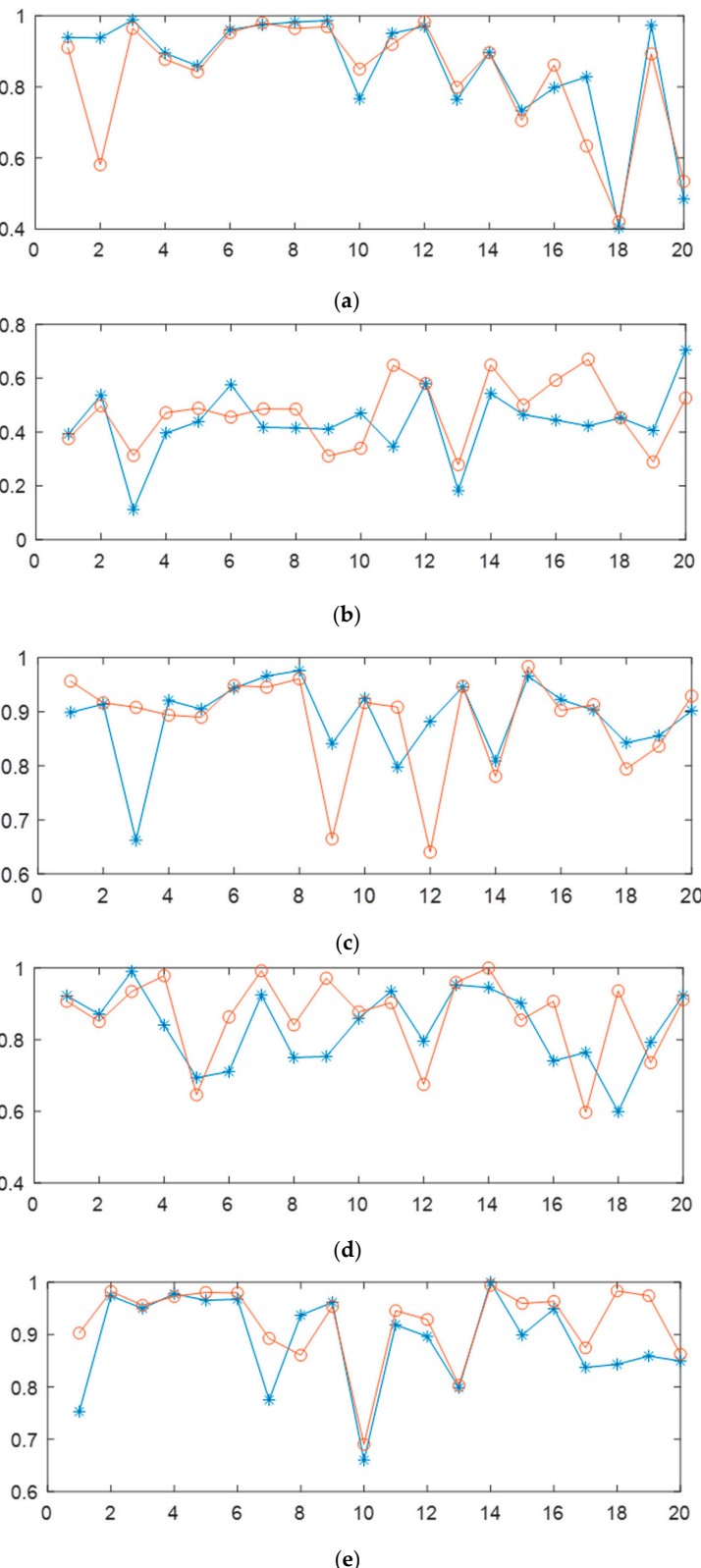

**Figure 8.** On the best network model, using Sleep-EDF Database Expanded, the effects of different EEG channels on automatic sleep stage classification are shown in the figure. The blue line represents the EEG channel Pz-Oz, while the red line represents the EEG channel Fpz-Cz. Panels (**a**–**e**) correspond to Wake, N1, N2, N3, and REM stages, respectively. The horizontal axis represents subject IDs, and the vertical axis represents accuracy.

## 6. Conclusions

This article proposes a novel RNN based on data bundling augmentation and label redirection for automatic sleep stage classification. The method uses data bundling augmentation to directly extract deep features of transition stages, which enhances its expression capability and effectively addresses the problem of misclassification in these stages. Additionally, the model employs label redirection combined with attention mechanism to significantly improve the classification accuracy of the N1 stage. The proposed model was evaluated on three datasets and demonstrated stable and good performance, which not only reflects the great potential of the network model in automatic sleep stage classification, but also validates its generalization ability and provides a solid foundation for exploring transfer learning capabilities in more datasets. Furthermore, this article also tests and analyzes the influence of different EEG channels on each sleep stage. Specifically, some channels have better recognition performance in certain sleep stages, while in other stages, the performance is not ideal. These findings contribute to a deeper understanding of the changes in EEG signals in different sleep stages, and improve the accuracy and reliability of automatic sleep stage classification, providing valuable data and references for further research on sleep problems.

**Author Contributions:** Conceptualization, Y.G. and Y.L.; methodology, C.L.; software, F.W.; validation, T.L., Y.G. and F.W.; formal analysis, C.L.; investigation, Y.L.; resources, Y.G.; data curation, Y.G.; writing—original draft preparation, F.W.; writing—review and editing, Y.G.; project administration, Y.G. All authors have read and agreed to the published version of the manuscript.

**Funding:** This work was supported by Jilin Province Changchun City Science and Technology Bureau Key R&D planning project of China under Grant No. 21ZGM25.

**Data Availability Statement:** No new data were created in this study. The datasets used in this article are all publicly available on the Internet and are accompanied by citation instructions as required by the database providers, which can be found in the references.

**Acknowledgments:** Thanks to Yudan Lv and Chang Liu for their medical knowledge support.

**Conflicts of Interest:** The authors declare no conflict of interest.

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
