# Peer review of "Automatic Sleep Staging Using BiRNN with Data Augmentation and Label Redirection"

_electronics, doi:10.3390/electronics12112394_

Round 1

Reviewer 1 Report

The research is on sleep staging where the authors proposed an automatic sleep stage classification model based on dual-layer RNN network and they evaluated on multiple public datasets which achieved high overall accuracy rate. Additionally, they investigated the impact of different EEG channels on the classification of sleep stages. The study has novelty, the write-up is good, however, it can be further improved.

The title includes the word Bidirectional Recurrent Neural Network (BiRNN) but in the text was dual-layer RNN (page 2, line 54 and line 57). But in page 4, line 163 and line 167, the words two-layer bidirectional recurrent neural network were used. Please standardise the term.

The abbreviation was first used in the title, then in the abstract and keywords. Then in the sub-title 3.3 at Page 4, line 160 and later at Page 5, line 181. Please provide the full word before using the abbreviation.

Page 1, line 37. Please provide the full word of REM before using the abbreviation.

Page 1, line 44: Please provide the full word of CNN, RNN, ANN before using the abbreviation.

Page 3, line 106. The use of EEG, EOG, and EMG abbreviations. Please provide the full word first.

Page 3 (Method). What is the specification of the computer or laptop used in this study?

Page 3, line 110. The use of PSG abbreviation. Please provide full word first.

Page 5, line 188. The use of LSTM and GRU abbreviations. Please provide full word first.

Page 7, line 218 and 220. The use of ST and SC abbreviations. Please provide full word first.

Page 12, line 360. The authors used Fig8 in the text, not standardise with the usage of Figure 1 – Figure 7 earlier on.

The write-up is good

Reviewer 2 Report

All notes and comments are included in the file attached to the review.
